# Exosomes Could Offer New Options to Combat the Long-Term Complications Inflicted by Gestational Diabetes Mellitus

**DOI:** 10.3390/cells9030675

**Published:** 2020-03-10

**Authors:** Juliana Ferreira Floriano, Gareth Willis, Francesco Catapano, Patrícia Rodrigues de Lima, Fabiana Vieira Duarte Souza Reis, Angélica Mercia Pascon Barbosa, Marilza Vieira Cunha Rudge, Costanza Emanueli

**Affiliations:** 1Botucatu Medical School, Sao Paulo State University, 18618687 Botucatu, Brazil; juliana.floriano@unesp.br (J.F.F.); patriciarl.lima@gmail.com (P.R.d.L.); fabianavdsreis@gmail.com (F.V.D.S.R.); angelicapascon@gmail.com (A.M.P.B.); 2Division of Newborn Medicine/Children’s Hospital, Harvard Medical School, Boston, MA 02115, USA; Gareth.Willis@childrens.harvard.edu; 3National Heart and Lung Institute, Imperial College London, London W12 0NN, UK; f.catapano@imperial.ac.uk

**Keywords:** gestational diabetes mellitus, outcomes, urinary incontinence, therapy, exosomes, microRNAs

## Abstract

Gestational diabetes Mellitus (GDM) is a complex clinical condition that promotes pelvic floor myopathy, thus predisposing sufferers to urinary incontinence (UI). GDM usually regresses after birth. Nonetheless, a GDM history is associated with higher risk of subsequently developing type 2 diabetes, cardiovascular diseases (CVD) and UI. Some aspects of the pathophysiology of GDM remain unclear and the associated pathologies (outcomes) are poorly addressed, simultaneously raising public health costs and diminishing women’s quality of life. Exosomes are small extracellular vesicles produced and actively secreted by cells as part of their intercellular communication system. Exosomes are heterogenous in their cargo and depending on the cell sources and environment, they can mediate both pathogenetic and therapeutic functions. With the advancement in knowledge of exosomes, new perspectives have emerged to support the mechanistic understanding, prediction/diagnosis and ultimately, treatment of the post-GMD outcomes. Here, we will review recent advances in knowledge of the role of exosomes in GDM and related areas and discuss the possibilities for translating exosomes as therapeutic agents in the GDM clinical setting.

## 1. Gestational Diabetes Mellitus

Gestational diabetes mellitus (GDM) is an increasingly common condition, affecting approximately 8.3% of pregnancies [1] worldwide. GDM occurs when insulin resistance exceeds the capacity for insulin secretion. The resulting insulin imbalance leads to vascular inflammation [2,3] and predisposes women to the risk of developing more severe pathologies [4].

Currently, the mechanisms underpinning GDM development are poorly understood, as well as the concomitant complications caused by a GDM pregnancy in mother and offspring. The risk of type 2 diabetes mellitus (T2DM) and cardiovascular diseases (CVD) rates, are rising alarmingly in the general population and is further increased for both mother and child after a GDM pregnancy [5,6,7]. Furthermore, for the mother, GDM is a strong predictor of urinary incontinence (UI) up to two years postpartum even in cases of cesarean section, where there is no vaginal distention, due to gestational diabetic myopathy [8,9,10].

UI dramatically diminishes women’s’ quality of life and represents a considerable economic burden for both patients and public health [11,12,13,14]. Hyperglycemia and reduced insulin signaling are deleterious for skeletal muscle cell metabolism and might indeed play a relevant role in GDM-associated pelvic muscle degeneration and atrophy [15,16,17,18,19,20,21,22]. Additional skeletal muscle changes leading to muscle weakness can result directly and/or indirectly from altered CCL7, relaxin, insulin, glucose, parathyroid hormone (PTH), calcium (Ca), calcitonin and vitamin D levels, chemokines, proteins and growth factors that can enact tissue homeostasis [23,24,25,26] and induce structural changes in skeletal muscle, decreasing the number of mitochondria, the functional capacity and leading to muscle weakness [27,28]. Additional GDM-related changes include hormones-related connective tissue remodeling that are still poorly understood in GDM [29]. There is no effective treatment for gestational diabetic myopathy. However, the treatment for UI is ineffective in a large proportion of the population, thus increasing public health costs, social spending and diminishing the quality of life of the affected women. Increased clarity on the pathways underlying GDM is therefore needed for preventing and minimizing GDM-associated manifestations [17,18,19,20,21,22].

## 2. Exosomes

Exosomes are small (~50–150 nm in diameter) extracellular vesicles (EVs), which are actively secreted by all cell types. They were accidentally discovered in 1983 by Rose M Johnstone and Bin-Tao Pan [30,31] whilst they were studying how iron enters maturing red blood cells. These first studies suggested their function as being an alternative to lysosomal degradation [32,33] allowing the discard of transferrin receptors, which had become useless in mature red blood cells [31]. At the same year Harding et al., 1983, found the same results suggesting that transferrin is internalized via coated pits and vesicles, they demonstrated that transferrin and its receptor are recycled back to the plasma membrane after endocytosis [34]. Since this inglorious debut as refuse clearance operators, exosomes have climbed the ladder of significance quite dramatically. Today, exosomes are recognized as important actors in cell to cell communication [32,33,35,36,37,38]. Several reports have shown that exosomes play important roles in a diverse array of physiological actions, including the immune response, tumor progression and neurodegenerative disorders [33,35].

Exosomes contain multifarious cargos including proteins, mRNAs and miRNAs and other cytosol components enclosed in a lipid bilayer [36,37,38]. They can shield their cargo content from enzymatic degradation. This ability is fundamental for intercellular communication. In fact, exosomes can shuttle their biologically active cargos from the parent cell to induce expressional and functional response in their recipient cells [39,40]. The modalities of exosomes-based communications potentially allow for the combination of multiple actions: exosomes released from the same MVBs could support pools of ligands able to engage different cell-surface receptors simultaneously, mimicking interaction between two cells but without the need for direct cell-to-cell contact. Exosomes binding to recipient cell membrane could also provide the beneficiary cells with ‘new’ surface molecules, permitting an increase in the range of cell targeting and potentially acquiring new adhesion properties [33]. Exosomes participate in the maintenance of normal tissue and cell physiology for example, stem cell maintenance [41], tissue repair [42], immune surveillance [43] and blood coagulation [44]. Exosomes have also been linked to pathogenic mechanisms in cancer [45,46,47], virus infection [48], neurological degenerative diseases [49] and pregnancy complications by GDM or preeclampsia [50,51,52,53,54].

Exosomes are released from a variety of cell types into the extracellular space [32,55,56] and are present in many biological fluids, including plasma, serum, amniotic fluid, urine and breast milk [46,57,58]. The concentration and content of circulating exosomes are potentially a rich source for novel clinical biomarkers and could additionally help in deciphering the mechanisms underpinning GDM complications, thus aiding in identification of new targets for therapeutic intervention [9,59,60,61]. On the other hand, as extensively described in this article, the discovery that exosomes mediate the wide-ranging therapeutic efficacy of stem cells, opens up new therapeutic avenues, which have relevance in the GDM area.

Arguably, as the result of a multidisciplinary and relatively ‘new’ research field, the precise nomenclature and classification of EVs remains troublesome [62]. The International Society for Extracellular Vesicles (ISEV) encourage the adoption of the term ‘EVs’ when referring to secreted vesicles [63,64,65]. In practice, EVs are frequently categorized into heterogeneous EV subsets such as exosomes, microvesicles or apoptotic bodies, based on their perceived route of biogenesis and biophysical characterization (for example, the vesicle diameter). The capacity to uphold one terminology over another remains challenging. In part, this is due to ‘crude’ EV isolation protocols that often co-isolate non-EV material. To confound this issue, such immature EV isolation methods are often accompanied by incomplete/ ‘poor’ EV characterization. Therefore, when interpreting EV studies, it is wise to exercise caution and consider the EV isolation method, the EV characterization and subsequent functionality/model/disease employed. EV biogenesis and nomenclature has been extensively reviewed [62]. Henceforth, for the purpose of inter-study interpretation, in this review we will adopt the nomenclature chosen by the cited articles where appropriate.

### 2.1. Further Particulars of Exosome Characteristics and Methodologies for Exosome Extraction for Fluid and Exosome Analyses

Exosomes are ~50–160 nM endocytic vesicles (enrichment in HSP 70, tetraspanins, Tsg101, Alix, Major histocompatibility complex (MHC) molecules) limited by a lipid bilayer and characterized by a defined density (flotation at 1.13–1.21 g/mL on a sucrose gradient) [33,66,67]. According to the current version of the exosome content database, ExoCarta (Version 4, http://www.exocarta.org), 4563 proteins, 194 lipids, 1,639 mRNAs and 764 miRs have been identified in exosomes from a variety of cell types [67,68,69,70]. The main components of exosome membranes are lipids and proteins, which are enriched with lipid rafts [32,55,71]. The exosomal lumen has numerous proteins [33], chemokines, such as CCL2, CCL3, CCL4, CCL5, CCL20 and nucleic acids, including mRNAs, miRs and other non-coding RNAs (ncRNAs) [40,67,72], reflecting both the condition and origin of the parent or producer cell [73]. Moreover, an in-depth characterization of proteins common to most exosomes shows that they express tetraspanin proteins CD63, CD9 and CD81 [32,55,56], other membrane-bound proteins and chaperones have also been shown [33,74].

Several methodologies are currently being used to isolate, quantify and validate the exosomes [75,76,77,78]. The optimal exosome isolation method depends upon the intended therapeutic use, route of administration, source material (e.g., milk, plasma, urine, cell culture) [79]. Exosome isolation and detection methods have been extensively reviewed in detail [80,81,82,83,84,85,86].

### 2.2. Exosomal microRNAs

Exosomes contain a wide range of small RNAs, particularly microRNAs (miRs) but also other forms of small non-coding RNAs (vaultRNA, tRNAs and miRs) [87] and specialized mechanisms are involved in their recruitment and loading to exosomes [88]. In a human, there is in excess of 2000 miRs. miRs mediate post-transcriptional gene silencing; to do this a miR mainly binds to the 3′- and untranslated region (3′-UTR) of a pool of target messenger RNA (mRNA) [89]. However, miRs can also bind to the mRNA 5′-UTR or open reading frame (ORF) regions [89,90]. The involvement of miRs in many biological activities has been well documented, including cell proliferation, cell differentiation, cell migration, disease initiation and disease progression [91,92,93,94,95]. In addition to being packed into exosomes or microvesicles, extracellular miRs can be loaded into lipoproteins [96,97] or bound by AGO-2 protein outside of vesicles [98]. All three modes of action protect miRs from degradation and guarantee their stability during their transportation in body fluids [98,99,100,101,102,103]. The role of miRs in exosomes is gaining increasing scientific attention. Conveying information via circulating EVs is deemed to be a third way of intercellular communication that is as essential as the cell-to-cell contact-dependent signaling and miR signaling via transfer of soluble molecules [104]. The role of exosomal miRs is of great importance in gene expression, demonstrating potential for new therapies and regenerative medicine [105,106]. Of note, viral miRs are also present in exosomes produced by infected mammalian cells, representing a new mechanism exploited by human tumor viruses to spread their attack [107,108]. This pathogenic mechanism is fascinating and suggests the possible use of exosomes for synthetic RNA-based therapies (discussed below).

### 2.3. Clinical Trials to Date, Using Exosome as a Therapy

Exosomes of autologous or allogenic origin have already been tested in different patient populations with reassuring results in term of safety. The exosomes safety profile depends on the cell from which they were derived [109,110,111,112,113,114]. Encouragingly, the recent clinical trials using cell-derived exosomes did not report serious adverse reactions thus far [112,113,115]. Escudier et al. reported results from the first exosome Phase I trial in 2005, highlighting the feasibility of large-scale exosome production and the safety of exosome administration [115]. More specifically, they used autologous exosomes pulsed with MAGE 3 peptides (of note, MAGE-3 gene is expressed in many tumors but it is silent in normal tissues and antigens encoded by MAGE-3). It therefore may be a useful target for specific anti-tumor immunization of cancer for the immunization of melanoma patients [115]. In the same year, Morse et al. [112], reported their study, which tested the safety, feasibility and efficacy of autologous dendritic cell (DC)-derived exosomes (DEX) loaded with the MAGE tumor antigens in patients with non-small cell lung cancer. They showed that the production of the DEX vaccine was feasible and that DEX therapy was well tolerated in patients with advanced cancer. Some patients experienced long term stabilization of the disease and activation of immune effectors. In 2008, Dai et al. reported the phase I clinical trial of the ascites-derived exosomes (Aex) in combination with the granulocyte–macrophage colony-stimulating factor (GM-CSF) in the immunotherapy of colorectal cancer (CRC), showing that the therapy was safe and well tolerated and induced a beneficial tumor-specific antitumor cytotoxic T lymphocyte (CTL) response [113]. In the Phase II clinical trial of the use of DEX in non-small cell lung cancer (NSCLC), they are testing the clinical benefit of γ-Dex (exosomes derived from IFN-γ-treated DC) as a maintenance immunotherapy in cancer patients at stage IIIB-IV, responding to or stabilized with, cisplatin-based chemotherapy [114]. The study has so far confirmed the capacity of Dex to boost the NK cell arm of antitumor immunity in patients with advanced NSCLC [116]. A Phase I clinical trial using DEX to treat advanced melanoma illustrated that Dex treatment enhanced the proportion and absolute number of circulating NK cells and restored the expression of a type II transmembrane receptor Natural Killer Group 2 member D (NKG2D), which is expressed on circulating T and NK cells [117], thus stimulating the MHC unrestricted NKG2D dependent cytotoxicity. These data provide a mechanistic explanation on how Dex may stimulate non-MHC restricted-anti-tumor effectors and induce tumor regression in vivo [118].

According to the www.clinicaltrials.gov website [119], there are currently 148 studies listed as clinical trials using exosomes as therapy and/or diagnosis, to study the pathophysiology of the disease and to predict and understand the outcomes after therapy [119,120]. From these listed studies, in total only twelve studies (summarized in Table 1) are using exosomes in innovative therapeutic approaches.

### 2.4. Exosomes as a Drug delivery System

As previously indicated, exosomes are a cell-free natural system for ferrying functionally active RNA between cells. Their membranes protect the RNA from degradation and supposedly contain recognition systems (not yet sufficiently elucidated) allowing them to target recipient cells. This process possibly inspired the idea that synthetic small RNAs, such as small interfering RNA (siRNAs) [133,134], can be delivered by exosomes mimicking the molecular mechanism of endogenous miRs transportation [71]. In support, Wahlgren et al. used plasma exosomes as gene delivery platforms to transfer siRNAs and silence MAPK in monocytes and lymphocytes [135]. Moreover, Wahlgren et al., 2012 and Alexander et al., 2015, showed that exosome-delivered exogenous miRs can re-program the cellular response to endotoxin, where exosome-delivered miR-155 enhances while miR-146a reduces inflammatory gene expression [135,136]. Overall, these studies provide evidence that exosomes have the potential to regulate inflammatory response. Expanding from this, exosomes can represent an efficient vehicle for acidic nucleic therapies [135]. Recombinant adeno-associated virus (rAAV) vectors are sized about 20 nm (hence can fit into exosomes) and are considered promising vectors for gene therapies for the cardiac and skeletal muscles [137,138]. rAAV became the first clinically approved gene therapy product in the western world [139]. In contrast to the near absence of a cellular immune response against rAAVs, clinical trials have shown that pre-existing neutralizing antibodies act against the naturally occurring AAV serotypes (presumably, a result of a prior infection with wild-type AAVs) in more than half the patients; this represents a significant obstacle to the broad application of AAV gene therapy [138]. Exosomes could help and overcome this issue. During rAAV production, a fraction of AAV vectors are embedded in exosome and microvesicles. The “vexosomes” (vector-exosomes) can outperform conventionally purified AAV vectors in transduction efficiency [140] and evade human neutralizing anti-AAV antibodies. Moreover, they can be modified to express a tissue/cell targeting peptide on the exosome external membrane, to improve delivery specific areas [139].

Further studies have shown that exosome-based drug delivery systems may provide unique advantages, including limited or no undesired immunogenicity when self-derived exosomes are used, greater stability in the blood due to evasion of complement and coagulation factors, efficient delivery of cargo into the cytosol of the target cell and possibly fewer off-target effects due to the natural tendency of exosomes to act on specific target cells [141,142]. An encouraging example is that the intranasal administration of curcumin-enriched exosomes (Exo-cur) led to rapid delivery of an exosome encapsulated drug to the brain that was selectively taken up by microglial cells to subsequently induce their apoptosis. These results demonstrate that this strategy may provide a non-invasive and novel therapeutic approach for treating inflammatory-related diseases, even in less accessible organs such as the brain [143]. To incorporate drugs into exosomes, different possibilities have been suggested [144]. Lipophilic small molecules were passively loaded into exosomes during co-incubation with exosomes [145,146].

MSCs are easily harvested from a large variety of human tissues including those that can often be considered ‘medical waste’ such as: adipose tissue [147,148], liver [149], muscle [150], amniotic fluid [151], placenta [152,153], umbilical cord blood [147], dental pulp [154,155], human ESC [156] and other sources. MSCs have shown a scalable ability to mass-produce exosomes, which is a highly desired attribute for conversion of MSC-exosomes into drug delivery vehicles [157]. MSCs therefore represent suitable cell candidates for the mass production of exosomes for drug delivery. Moreover, there are also alternative sources for large-scale production of drug-enriched exosomes with low host immunogenicity. For example, bovine milk exosomes were shown to increase oral bioavailability, improve drug efficacy and safety and exhibit tolerance between species without adverse immune and inflammatory response [158].

In summary, exosomes represent an innovative and very promising drug delivery system and exosome -based therapeutics offer new hopes to satisfied unmet clinical needs.

## 3. Exosomes and miRs in GDM

As a consequence of the diverse functions that exosomes undertake, they can be an extremely important tool not only for studying the pathophysiology of GDM but also for a safer, more effective and personalized potential therapy. In this section we will highlight the advances in exosomes studies in GDM and the perspectives for their use in GDM biomarkers and therapy.

Saker et al. showed that the plasma concentration of exosomes is higher in normal pregnant women than in non-pregnant women [159] while a later study from Salomon et al. demonstrated that placental exosomes are released into the maternal circulation at the beginning of 6 weeks of gestation [160]. Interestingly, changes in maternal plasma exosome concentration compared with normal pregnancy has been detected in GDM women [160,161]. Nakahara et al. [162], investigated the profile of placental-derived exosomes (PdE) in a stratified cohort study based on normal (n = 30), GDM (n = 10) and preeclampsia (PE) pregnancy (n = 15) outcomes. They found that significant factors contributing to total variations of the PdE were gestational age and pregnancy outcomes, PdE levels increased in all types of gestation but in pregnancies complicated by GDM and PE these levels were significantly higher than in normal pregnancies. In addition, maternal body mass index (BMI), glucose concentration and fetal body weight significantly correlated with the concentration of PdE across gestation, suggesting that exosomes may be involved in maternal metabolic adaptation to pregnancy and therefore that PdE may be used as early predictor of adverse outcomes, including GDM and PE [162]. Salomon et al. described that PdE released from women with GDM may alter maternal physiology by a process of exosomal placento-maternal transfection a “payload” of receptors, proteins and/or oligonucleotides, that have been specifically pre-conditioned by the GDM placenta. The authors propose that some mediators act in this system, including the vascular, pancreatic and adipose tissues and innate immune response system [163]. They also describe paracellular effects associated with GDM mediated by trophoblasts or placental mesenchymal stem cells (MSCs), altering for example the endothelial activity promoting changes in transport glucose GLUT 3 and thus delivery of energy substrates to the fetus [163]. In addition, PdE may contribute to the proinflammatory state associated with pregnancy, an increased phenomenon under diabetic conditions [26,52]. Jayabalan et al. by using a bioinformatic analysis tool named Sequential Windowed Acquisition of All Theoretical Mass Spectra [SWATH] showed that exosomal proteins are primarily associated with energy production, inflammation and metabolism [164], major pathways compromised by GDM. The data may be of utility in elucidating the underlying physiological mechanisms associated with insulin resistance in GDM [165]. Another study suggests that in GDM, exosomes secreted from adipose tissue regulate placental glucose metabolism by improving the communication of adipose tissue derived exosomes (exo-AT) to placental tissues, this therefore might become an effective intervention strategy to prevent the consequences of GDM, such as fetal overgrowth [66,164]. Novel findings indicate that the insulin resistance (IR) observed in obesity is maintained by adipose tissue by releasing exosomes promoting IR and other obesity-related metabolic conditions [166]. These findings support the hypothesis that dysregulated secretion of adipose tissue-derived exosomes plays a pivotal role in the development of GDM in obese mothers [166]. To complement clinical investigations, in vitro studies demonstrated that high D-glucose increases the release of exosomes from first trimester trophoblast cells cytokines secretion, such as interleukin-8 (IL-8) and TNF-a from human umbilical vein endothelial cells (HUVECs), which are of fetal origin [51,167].

Importantly, Salomon et al., 2016, reported that higher concentrations of placental exosomes during early pregnancy (i.e., 11–14 weeks) are predictive of GDM [161] and that the circulating levels of both placental-derived and total exosomes are higher compared with normal pregnancy. However, the ratio between Placental Alkaline Phosphatase (PLAP)/placenta and total exosomes was decreased in the circulation of GDM mothers, which could be explained by the increased release of exosomes from non-placental sources [52,161]. Further studies are needed to investigate exosomes released from non-placental sources such as skeletal muscle and adipose tissue [161].

The study of exosomes as paracrine vectors might generate new knowledge for deciphering GDM pathophysiology for mother and fetus as well as providing precious biomarkers for the prediction and monitoring of the disease. Recently, studies on exosomal miRs provided an opportunity for a better understanding of the molecular processes of skeletal muscle diseases [168,169,170,171,172,173,174,175]. Research suggests that miRs play important roles in skeletal muscle development and several miRs have been identified as biomarkers for myogenesis, muscle mass changes and nutrient metabolism in physiological and pathological states [26,176,177,178]. Nair et al., 2018 found that placental exosomes in GDM carry a specific set of miRs associated with skeletal muscle insulin sensitivity [26]. The expression of this set of specific exosomal miRs, varied in a consistent pattern in the placenta, in circulating exosomes and in skeletal muscle in GDM. Placental exosomes from GDM pregnancies decreased insulin-stimulated migration and glucose uptake in primary skeletal muscle cells obtained from patients with normal insulin sensitivity. Interestingly, placental exosomes from NGT increase glucose uptake in response to insulin in skeletal muscle from diabetic subjects. These findings suggest that placental exosomes may have a role in modifying insulin sensitivity in normal and GDM pregnancies [161]. These results pave the way for a better understanding of gestational diabetic myopathy and UI in this population with GMD and for clarifying relevant aspects about insulin interaction with the muscle and muscle function.

Several studies suggest that miRs are involved in processes that contribute to the development and evolution of GDM. Dicer and Drosha are important for the miR biogenesis. Rahimi et al. found the dysregulation of Drosha, Dicer in pregnant and GDM patients when compared to healthy controls. They hypothesized that miRs are involved in the development of GDM [179]. Wander et al. found that circulating early–mid-pregnancy miRs are associated with GDM, particularly among overweight/obese women who are pregnant with male offspring [180]. Pillar et al. showed that miRs are involved in the pathogenesis of preeclampsia and GDM and have potential as early biomarkers for disease development [54]. Cao et al. concluded that plasma mRNA-16-5p, -17-5p and -20a-5p are potential diagnostic biomarkers in GDM [181]. Li et al. identified a miRNA signature involvement in GDM which may contribute to macrosomia through enhancing epidermal growth factor receptor (EGFR) signaling [182].

Another study demonstrated that the increase in the placenta-enriched miR (miRNA-518d) may contribute to the pathology of the development of GDM, via an effect on the regulation of proliferator-activated receptor-α (PPARα) expression [183]. miRs from adipose tissue, such as miR-222, might be a candidate biomarker and therapeutic target for GDM, due to the potential regulation of ERα expression in estrogen-induced insulin resistance in GDM [184]. Exosomal miRs can be profiled in biomarker discovery studies [185]. Trophoblast/cytotrophoblast cells compose the placenta. Consequently, the miRs contained in PdE might help to better define the mechanisms underlining fetal-maternal interaction [77,186] as well as to study placental dysfunction. These PdE-miRs might play an important role in GDM pathogenesis. Therefore, via the study of these exosomal miRs, important aspects could be better understood and explored for more effective future diagnoses and new therapeutic approaches to GDM [53].

## 4. Novel Therapeutic Approaches in Gestational Diabetes Mellitus

During the past decade, stem cell therapy studies have focused on the use of multipotent adult stem cells, particularly mesenchymal stem/stromal cells (MSC), which we will discuss in more depth below. These studies have highlighted new therapeutic promises for a treatment of a variety of conditions including UI and CVDs [187,188,189,190]. However, the clinical success of cell therapy has not yet been confirmed in large human studies. Furthermore, animal studies have delivered the knowledge that beneficial pro-regenerative and anti-inflammatory effects of stem cell therapies are mediated by a paracrine action and/or by acute immune response to cell delivery rather than an in situ transdifferentiation [191,192].

MSCs can be extracted from different sources, including the bone marrow, amniotic fluid [151] and placenta [193] and possess tissue protective and regenerative attributes together with immunomodulatory, anti-inflammatory, proangiogenic and antifibrotic capacities. Recent studies have established that one of the main therapeutic vectors of MSCs is represented by EVs, particularly exosomes [194]. However, the mechanisms by which MSC-exosomes afford their beneficial actions remain incompletely understood. Several studies suggest that MSC-exosomes regulate immune responses [195,196]; reinforcing this proposition, recent data suggests that macrophage (MΦ) immunomodulation is the ‘gatekeeper’ to the success of MSC-exosome therapies [197,198]. MSC-exosomes reportedly modulate MΦ phenotype, suppressing the proinflammatory M1-like state and shifting the M2-like MΦ to favor an anti-inflammatory, pro-regulatory phenotype both in vitro and in vivo [197,198]. However, exosomes extracted from cells cultured in cardiometabolic disease [199,200,201,202,203,204,205] mimicking conditions or from biological fluids of diabetic patients, produce pathogenic microangiopathic effects [201,202]. Taken together, extreme caution should be exercised when considering the use of ‘naïve’ patient-derived exosomes for therapeutic intervention. In contrast, several reports indicate that allogenic “healthy” MSCs are an excellent source of bioactive exosomes [203], endowed with protective, antifibrotic and proangiogenic properties [204]. Additionally, MSCs reportedly recruit CCR2+ monocytes (Mo), which were shown to contribute to the regenerative properties induced by stem cell injection into the rodent heart [192,205]. Although great success has already been achieved using cell therapy with mesenchymal stem cell (MSC), many aspects related to their effectiveness and side effects need to be deciphered, including their potential carcinogenicity [79,206,207] as well as their migratory ability (to different sites) and the resulting engraftment potential [208,209,210]. These challenges become even bigger in GDM treatment through stem cell therapy, as there is a major limitation on the safety of therapy in this case, to avoid deleterious effects that might reach the conceptus.

For these reasons new acellular approaches to therapy have been employed in recent years as a basis for the knowledge gained about on the paracrine action of MSCs through exosomes and their important biological role in the body [207,211].

Therapies based on MSC-exosomes represent a promising scientific field to be tested for the capacity to improve the post-GDM outcome of mother and child. Exosomes have a good potential for protective and regenerative therapies and in GDM-caused myopathy, they could be delivered locally for example by (co-injection, mixing with hydrogels or coating scaffolds with exosomes using fibrin gels or specific linkers) [212].

Interestingly, human adipose-derived stem cell (hADSCs)-exosomes showed some promise to contrast stress urinary incontinence (SUI), a common medical condition affecting approximately 30% of postpartum women [213]. Specifically, Ni et al. found that administration of hADSCs-exosomes provided functional and histological improvements in a rodent model of SUI [213]. In addition, they found hADSCs-exosomes harbored several proteins associated with PI3K-Akt, Jak-STAT and Wnt signaling pathways, that were associated with skeletal muscle and nerve regeneration and proliferation improving the SUI [213]. In accordance, Liu et al. found that hADSCs-exosomes increased type I collagen content by increasing collagen synthesis and decreasing collagen degradation in vaginal fibroblasts from women with SUI, supporting the notion that these exosomes may be a novel therapeutic approach for treating SUI [214]. More recent work from the same research group also showed that exosomes secreted by fibroblasts from women with SUI play an important role in regulating endothelial cell angiogenesis [215].

Interestingly, Wu et al. showed that exosomes derived from stem cells contained in the urine (USCs-Exo) can improve skeletal muscle regeneration in pubococcygeus muscle injury in rats [216]. Here, the authors found that USC-Exos act on satellite cells promoting their activation, proliferation and differentiation via an enhancement of the phosphorylation of extracellular-regulated protein kinases (ERK). This paper has therefore identified a novel agent for skeletal muscle regeneration providing a basis for further exploring a cell-free correction for SUI [216]. Notably, GDM is associated with other comorbidities such as gestational hypertension (HTN), hypothyroidism, obesity and lipid abnormalities, which can become chronic throughout the woman’s life [217,218,219]. On balance, exosomes hold promise as a novel therapeutic approach for pathologies associated with GDM.

The use of exosomes as therapy for GDM outcomes, such as UI and CVD are still premature but encouraging in vitro and preclinical results provide promise. Table 2 shows an overview of the potential of exosomes in protective regenerative medicine in the context of GDM [212,220].

## 5. Concluding Remarks

Once the scientific community has fully succeeded in harnessing the beneficial properties of exosomes, unlocking their potential for drug delivery and the correction of gene expression in specifics cells and tissues, they could become powerful and sophisticated tools, in the emerging field of nanomedicine (summarized in Figure 1). Improving fundamental knowledge on exosome structure, biogenesis, roles in cell-to-cell signaling, the process of recognition and internationalization of exosomal content all combined, should allow further improvement of the methods to manipulate the exosome cargo and then deliver it in a very precise and effective way. However, as illustrated above, there are still a number of important challenges to be addressed, such as dosage and route of application to obtain desired therapeutic effects, tracking of exosomes in the body in target cells or tissues, as well as the long-term evaluation of side effects that therapy may cause. Investigation of the maternal plasma exosomes concentration and molecular content might add important knowledge to the understanding of the GDM etiology and complications on mother and child and then be translated into diagnostic and predictive biomarkers in a clinical context.

## Figures and Tables

**Figure 1 cells-09-00675-f001:**
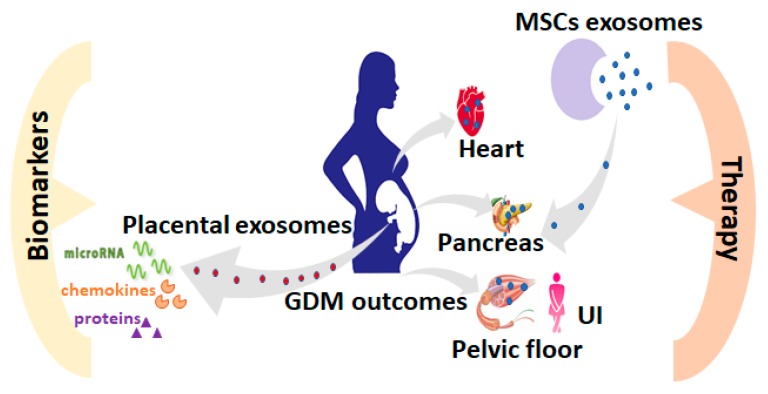
Overview of the potential impact of exosome in GDM. The therapeutic approach is shown on the right side: “Therapeutic” exosomes (from mesenchymal stem/stromal cells or alternative sources) could be used to reduce the negative impact of GDM on cardiovascular disease (exemplified by the heart in the figure), diabetes mellitus (simplified by the pancreas) and on the myopathy of pelvic floor and rectus abdominis muscle, which contribute to urinary incontinence (UI) up to 2 years postpartum. The diagnostic approach is shown on the left side: Through the exosomal content profile, new biomarkers of GDM onsets (diagnostic biomarkers) and post-GDM negative outcome (diagnostic and prognostic biomarkers) could be developed to improve the treatment of the GDM and the outcome for the mother and child.

**Table 1 cells-09-00675-t001:** Current clinical trials testing the therapeutic potential of exosomes from several sources in different human diseases.

Exosomes Type	Condition to Be Treated	Locations	Reference
CAP-1002 (Cardiosphere-Derived Cells: CDCs)	Duchenne muscular dystrophy	Multicenter American Study (California, Florida, Missouri, Ohio, Utah, Wisconsin)	[121]
Curcumin conjugated with plant exosomes	Colon cancer	University of Louisville, USA	[122]
Ginger and aloe plants exosomes	Polycystic ovary syndrome	University of Louisville Louisville, Kentucky, USA	[123]
DEX	Cancer vaccination to lung cancer	Gustave Roussy, Cancer Campus, Grand Paris	[124]
MSC-derived exosomes with KrasG12D siRNA (“iExosomes”)	Pancreatic cancer	M.D. Anderson Cancer Center, USA	[125]
MSC^Exo^	Healing of large and refractory macular holes	Tianjin Medical University Eye Hospital (China)	[126]
MSC derived microvesicles and exosomes	Type I Diabetes Mellitus	General Committee of Teaching Hospitals and Institutes, Egypt	[127]
Umbilical mesenchymal stem cells derived exosomes	Dry eye symptoms in patients with chronic Graft Versus Host Diseases (cGVHD)	Zhongshan Ophthalmic Center, Sun Yat-sen University, China.	[128]
Exosomes derived from amniotic liquid stem cell	Depression, anxiety and dementia	Neurological Associates of West Los Angeles, USA	[129]
Exosome produced from neonatal stem cell	Craniofacial neuralgia	Neurological Associates of West Los Angeles, USA	[130]
MSC^Exo^ enriched by miR-124	Disability of patients with acute ischemic stroke	Isfahan University of Medical Sciences, Iran	[131]
Stem cell conditioned medium	Chronic ulcer wounds [12]	Stem Cell and Cancer Institute, Kalbe Farma TbkPT Pharma Metric Labs, Indonesia.	[132]
Human MSC-exosomes	Bronchopulmonary dysplasia	United Therapeutics USA.	[119]

**Table 2 cells-09-00675-t002:** Overview of the characteristics of mesenchymal stem cell (MSC)^exo^ that suggest their potential in gestational diabetes mellitus (GDM) treatment.

Biological Process	Effects	Reference
Angiogenesis/Cell proliferation	Proliferation, migration and tube formation of endothelial cells through the Wnt4/β-Catenin Pathway/ Transferring miR), tube formation into endothelial cells miR-135b and by targeting factor-inhibiting HIF-1/ Promotes the enhancement of the proliferation and migration of fibroblasts by transferring signals to target cells activating several signaling important pathways (Akt, ERK and STAT3) and inducing the expression of a number of growth factors - [hepatocyte growth factor (HGF), insulin-like growth factor-1 (IGF1), nerve growth factor (NGF) and stromal-derived growth factor-1 (SDF1)]/ Inducing neovascularization in preclinical models by the paracrine effect by transferring pro-angiogenic microRNAs / Endothelial cell angiogenesis by transferring miR-125a/	[221,222,223,224,225]
Immunomodulation	Immunomodulatory effect of human stimulated T cells by inhibitory effect in the differentiation and activation of T cells as well as a reduced T cell proliferation and IFN-γ release/ Modulation of the local and systemic maternal immune system by exosomes secreted from trophoblast cells that carry HLA-G and B7 family immunomodulators/ MSC-derived exosome possesses the immunomodulatory properties mediated by paracrine factors suppressing the secretion of pro-inflammatory factor TNF-a and IL-1b, increasing TGF-β, inducing the conversion of T helper type 1 into T helper type 2 also reducing the potential of T cells to differentiate into IL 17/ Exosomes are the trigger the release of cytokines/chemokines from immune cells and stimulation of anti-tumor immune reactions or in a systemic immunosuppression by inducing the secretion of pro-inflammatory cytokines such as IL-1β, tumor necrosis factor (TNF)-α, IL-23a, CCL5 (RANTES) and IL-6/ Exosomes from MSCs ameliorate experimental bronchopulmonary dysplasia and restore lung function through MΦ immunomodulation by suppressing the pro-inflammatory “M1” state and augmenting an anti-inflammatory “M2-like via Cytokines, such as Ccl2, Ccl7 and IL6/ MSC exosomes enhanced the survival of allogenic skin graft in mice by induced polymyxin-resistant by activating APCs via MyD88-dependent.	[198,226,227,228,229,230]
Tissue regeneration	Fibroblast activation to initiate tissue regenerative responses by delivering TGF-b1 mRNA among others yet to be identified moieties/ Osteochondral regeneration by the action of regulatory components including miRs, mRNAs and proteins/ Accelerate skeletal muscle regeneration by enhancing myogenesis and angiogenesis, which is at least in part mediated by miRs such as miR-494/ Enhance cartilage tissue regeneration and prevent osteoarthritis of the knee in a rat model by the overexpression miR-140-5p/ As biomimetic tools for stem cell differentiation inducing stem cell differentiation and tissue regeneration by signaling mechanisms triggered (P38 mitogen activating protein kinase pathway) from exosomes.	[231,232,233,234,235]

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
