# Peer review of "Exosomes Could Offer New Options to Combat the Long-Term Complications Inflicted by Gestational Diabetes Mellitus"

_cells, 2020, doi:10.3390/cells9030675_

Round 1

Reviewer 1 Report

This is an interesting work, reflecting the potential use of exosomes as therapeutic agents for management certain diseases. However, as the title mentions, the principal objective of the work is referred to gestational diabetes mellitus and its complications. In the first part of the paper, authors describe urinary incontinency as one of the complications after GDM. However, there is no much data about specific use of exosomes as therapeutic agents for this complication. I suggest to complement the epigraph 4 with more relevant data for GDM complications, including UI.

Reviewer 2 Report

The review by Floriano et al. describes the possibilities of exosomes as diagnostic or therapeutic agents for gestational diabetes Mellitus (GDM). The authors covered recent advances in knowledge of the role of exosomes in various diseases including GDM as well as current methodologies for exosome extractions and analyses. Focussing GDM and exosomes could be a unique topic and may attract a broad audience.  Major comment 1: The authors only mentioned the roles of exosomes, not other extracellular vesicle (EV) subpopulations, and the rationale behind it was not clearly stated. There are various technological limitations in EV extraction techniques and accurate analysis of EV subpopulations. Without mentioning this, this review may give the reader a more positive impression on exosomes among other EV subpopulations. Some of the references contain the data from non-exosome subpopulations. References need to be carefully reassessed if the role of exosomes in GDM is focussed. Major comment 2: many of the landmark studies are missing in the current reference list. It was described that exosomes were accidentally discovered in 1983 by Rose M Johnstone and Bin-Tao Pan. However, in the same year, Harding et al. independently reported “Receptor-mediated endocytosis of transferrin and recycling of the transferrin receptor in rat reticulocytes.” In addition, Melo et al. (ref. 34) was cited in the introduction of exosomes, although this report has been provocative in the field. Generally,  the review articles inform readers well established and widely accepted ideas. The introduction of exosomes needs to be reconsidered to include more landmark studies such as Valadi et al. (2007), Skog et al. (2008), AI-Nedawi et al. (2008), or Peinado et al. (2012).  Major comment 3: If the focus should be on the roles of exosomes in GDM, the methodologies for exosome extractions and analyses (Table I) seems a bit disjointed. In this context, I would argue against the significance of the table since it seems a bit out of the scope of the review and it does not cover all techniques, for example, the microfluidics-based device only mentioned the acoustic wave-based isolation technique. As ref. 80 describes progress in these techniques very well although there might be newer review articles, citing other excellent reviews would be enough here.  Major comment 4: Table 3 should provide more specific information such as miR types, proposed mechanisms, or molecules. In addition, fig. 1 only indicates miR-based biomarkers. Since exosomes carry a variety of molecules, it would be better to include that impression in the image. Minor comment 1: I don’t understand the significance and relevance of the paragraph regarding plant-derived exosomes. Minor comment 2: Ref. 105, 106, 107, clinical studies of EVs should include original research papers such as Escudier et al. J. Transl. Med. (2005), Morse et al. J. Transl. Med. (2005), or Kordelas et al. Leukemia (2014). Minor comment 3: Ref. 42 is not relevant to blood coagulation; transmission microscopy should be transmission electron microscopy; placenta-enriched mRNA (miR-518d) should be miRNA or microRNA. All the spelling errors need to be corrected. 

Round 2

Reviewer 1 Report

Accept in the present form